# Identification of the Keratin-Associated Protein 22-2 Gene in the *Capra hircus* and Association of Its Variation with Cashmere Traits

**DOI:** 10.3390/ani13172806

**Published:** 2023-09-04

**Authors:** Zhanzhao Chen, Jian Cao, Fangfang Zhao, Zhaohua He, Hongxian Sun, Jiqing Wang, Xiu Liu, Shaobin Li

**Affiliations:** 1Gansu Key Laboratory of Herbivorous Animal Biotechnology, Faculty of Animal Science and Technology, Gansu Agricultural University, Lanzhou 730070, China; m18893492402@163.com (Z.C.); zhaofangfang@gsau.edu.cn (F.Z.); hezh@st.gsau.edu.cn (Z.H.); 18089350617@163.com (H.S.); wangjq@gsau.edu.cn (J.W.); liuxiu@gsau.edu.cn (X.L.); 2Faculty of Bioengineering, Jiuquan Vocational Technical College, Jiuquan 735000, China; caojian@st.gsau.edu.cn

**Keywords:** keratin-associated protein *KAP22-2* gene (*KRTAP22-2*), variation, *Capra hircus*, mean fiber diameter, in situ hybridization, non-synonymous mutation

## Abstract

**Simple Summary:**

Keratin-associated protein (KAP) and keratin intermediate filament protein (KIF) are the main structural proteins that make up cashmere fibers. It has been shown that keratin-associated protein composition and genetic diversity are closely related to fiber quality. In this study, we identified a new member of the goat *KRTAP* family, *KRTAPP22-2*. Four specific bands (A-D) were detected in the Longdong Cashmere goat population, forming six banding patterns individually or in combination. Sequencing detected four single nucleotide polymorphism sites (SNPs), distributed in both coding and non-coding regions, and all were non-synonymous SNPs. In addition, an insertion sequence of 6-bp length was present in allele C, resulting in a two-amino acid insertion. A population of 356 Longdong Cashmere goats was used to analyze the association between the *KRTAP22-2* gene and cashmere traits. After excluding alleles with frequencies less than 5%, the results showed that the mean fiber diameter of goats from the AB genotype was significantly higher than that of the AA and AC genotypes. This suggests that the newly identified *KRTAP22-2* gene may have a greater role in goat breeding for reducing cashmere fiber diameter.

**Abstract:**

The Cashmere goat is an excellent local goat breed in Gansu Province of China, and it is expected to improve cashmere production and cashmere quality through selection and breeding to enhance its commercial value. Keratin-associated proteins (KAPs) play an important role in maintaining wool structure. The gene encoding the keratin-associated protein 22-2 (*KAP22-2*) gene has been identified in selected species other than goats, such as humans, mice, and sheep. In this study, the sequence of the sheep *KAP22-2* gene (*KRTAP22-2*) was aligned into the goat genome, and the sequence with the highest homology was assumed to be the goat *KRTAP22-2* sequence and used to design primers to amplify the goat gene sequence. A total of 356 Longdong Cashmere goats (Gansu Province, China) were used for screening of genetic variants. Four specific bands were detected by polymerase chain reaction-single-stranded conformational polymorphism (PCR-SSCP) analysis, and they formed a total of six band types individually or in combination. Four alleles were identified by DNA sequencing of PCR amplification products. A total of four single nucleotide polymorphic sites (SNPs) were detected in the four sequenced *KRTAP22-2* alleles. Two of them are in the 5’UTR region and the other two are in the coding region, and the variants in the coding region are all non-synonymous mutations. In addition, there was a 6 bp length variation in allele C. The gene was expressed in the cortical layer of primary and secondary hair follicles, the inner root sheath, as well as hair papillae and hair maternal cells in goats. The results of the correlation analysis between genotypes and cashmere traits showed that after excluding genotypes with a gene frequency of less than 5%, the mean fiber diameter (MFD) of cashmere was significantly higher in the AB genotype than in the AA and AC genotypes. That is, the *KRTAP22-2* gene variants are associated with mean fiber diameter in cashmere. The above results suggest that the goat *KRTAP22-2* variant can be utilized as a molecular marker candidate gene for cashmere traits.

## 1. Introduction

Cashmere is currently the only foreign exchange-generating livestock product that can influence the international market in China. In recent years, the number of goat farming has also gradually increased, although individual cashmere production has significantly improved, the selection and breeding of cashmere quality has been neglected, resulting in slow growth of the fleece and coarsening of the cashmere fineness, which has resulted in low economic benefits of cashmere goat farming. Improving the individual cashmere quality of goats through selection and breeding is an effective way to guarantee the economic income of farmers, and is of great practical significance. The Longdong Cashmere goat is a representative breed of cashmere goats in Northwest China, but its industrial development has encountered the same problems as other cashmere goat varieties.

Of all the traits of cashmere, weight and average fiber diameter are important traits that reflect economic value [1]. Keratin-associated proteins (KAPs) are important structural components of wool [2], which crosslink with keratin (K) in keratin intermediate filaments (IFs) [3]. The KAPs are complex proteins that generally include a variety of amino acids such as cysteine, tyrosine, etc. [4]. The KAPs are encoded by a gene family (*KRTAPs*), which are in clusters on chromosomes [5]. They are classified as high-sulfur KAPs, ultra-high-sulfur KAPs, and high-glycine-tyrosine KAPs [6]. All human’s *KRTAPs* have been identified in sheep and goats except for *KRTAP25-1* [7]; however, *KRTAP8-2* [8,9] has been found in sheep and goats, which has not been identified in humans. This suggests that sheep and goats have more genes encoding KAPs.

More than 100 KAP genes have been identified in different species and are classified into different families according to their characteristics [10,11,12]. The 17 functional KAP genes in humans form seven HGT-KAP families [10,11,12], and fewer HGT-KAP genes have been reported in caprine compared to humans, with only five caprine HGT-KAP genes from three families reported. These caprine HGT-KAP genes are *KRTAP7-1*, *KRTAP8-1*, *KRTAP8-2*, *KRTAP20-1*, and *KRTAP20-2*. All of the KAP genes have no introns, consist of only a single exon, and have high levels of cysteine or high levels of glycine and tyrosine [3]. In HGT-KAPs, the special structure of tyrosine gives it a freer spatial structure of its own and may form a hydrogen bond with the benzene ring core of another aromatic amino acid [13], which would allow for stronger ring layer interactions and may lead to further strength in wool fibers with a degree of flexibility [14]. Many *KRTAPs* have abundant nucleotide repeats [15,16] and high GC content, which leads to reduced DNA polymerase activity and increased DNA polymerase slip, resulting in higher mutation rates [17].

KAP22-2 is an HGT-KAP that has not yet been reported in goats. In this study, the sequence of the putative goat *KRTAP22-2* gene was identified, and DNA sequencing and other techniques were also used to reveal the chromosomal localization, variation, and amino acid composition of the gene, and to analyze the relationship between cashmere traits and gene variation in Longdong Cashmere goats. The expression of the *KRTAP22-2* gene in goat fiber follicles was detected using in situ hybridization, and the effect of the gene on the physicochemical properties of cashmere was Investigated from the histomorphological point of view.

## 2. Materials and Methods 

Experiments with animals were approved by Gansu Agricultural University, and the Ministry of Science and Technology of the People’s Republic of China has developed Guidelines for the Breeding and Use of Laboratory Animals. All goat experiments were conducted in accordance with the guidelines (approval number 2006-398).

### 2.1. Animals, Geographic Location, and Growing Conditions

The experimental animals were Longdong Cashmere goats which were kept in Huanxian County, Qingyang City, Gansu Province, China (between 36°1′–37°9′ N latitude and 106°21′–107°44′ E longitude), at an average altitude of 1426 m above sea level. These animals are grown and bred at the Yusheng Cashmere Goat Breeding Company.

### 2.2. Blood and Cashmere Samples of Goats

A total of 356 Longdong cashmere goats with a similar age were selected to study the genetic variation of *KRTAP22-2*. Blood samples from goats were collected from the neck and genomic DNA was purified using a two-step procedure described by Zhou et al. [18]. Collection of cashmere from the central region of the posterior margin of the left scapula in goats of about one year of age and measurement of its phenotypic characteristics.

### 2.3. Tissue Samples

During the anagen phase (October) and the telogen phase (March) of hair follicle development, six Longdong Cashmere goat rams that were all 3 years old were selected, with the same breeding conditions and not related by direct blood between them, and skin tissues were collected from the posterior border of the left scapula of these goats. The collected skin tissue was cut into centrifuge tubes, placed in liquid nitrogen for RNA extraction. The remaining samples were completely unfolded and placed in two slides for fixation, and then stored in the in situ hybridization solution for backup.

### 2.4. Search for a Caprine Homologue of Sheep KRTAP22-2 in the Goat Genome

BLAST homology search of the goat genome (NC_030808.1) with reference to the sheep *KRTAP22-2* gene sequence (NM_181615). The comparison results showed that a DNA sequence with 100% similarity to sheep *KRTAP22-2* was present on goat chromosome 1. This genomic sequence was assumed to be nominally goat *KRTAP22-2*. This sequence has been used for the design of PCR primers for the amplification of the target gene in goat DNA.

### 2.5. Primer Design and Amplification

The primer sequence information was: 5’-TGAGTCTAGCAGTGCCTGTG-3’ (3522275 to 3522256) and 5’-GATCCTCATAAAAGAACATCC-3’ (3521979 to 3521959), with the nucleotide positions referring to goat genome (NC_030808). These primers were synthesized by Sangon Biotech Co., Ltd. (Shanghai, China). PCR amplification was performed using a conventional 20 μL reaction system, which was stored at low temperature after a normal process including pre-denaturation, denaturation, annealing, and extension.

### 2.6. Screening for Variation in Caprine KRTAP22-2

The PCR amplicon sequence variants were screened by the SSCP method. The PCR amplification products were denatured in denaturant (98% formamide, 10 mM EDTA, 0.025% bromophenol blue and 0.025% xylene cyanide) at 105 °C for 10 min to denature the DNA to a single-stranded state. Then we cooled down the sample by placing it on the ice-water mixture prepared in advance, and took 6–10 μL of denaturing mixture into the wells of the polyacrylamide gel, and electrophoresis was initiated. The optimal SSCP electrophoresis conditions were 18 °C at 180 V for 16 h. Silver staining was performed according to Byun et al. [19].

### 2.7. Sequencing and Analysis of Allelic Variants

The genotype of the sample was determined to be homozygous or heterozygous based on the staining of the polyacrylamide gel. If the samples are homozygous, the PCR amplification products are sequenced directly. If the sample is genotypically heterozygous, the alleles are sequenced using the method described by Gong et al. [20]. Two-way sequencing by Shanghai Bioengineering Co. (Shanghai, China) The cutting colloid sequencing method is as follows: the polyacrylamide gel to be sequenced was washed with pure water and placed on a white background plate. The target band was carefully cut off with a scalpel blade, placed in an EP tube, washed with 1000 μL of ultrapure water, and the wash solution was aspirated with a pipette, then mashed three times. Then 200 μL of distilled water was added and placed in a water bath at 75 °C for 3 h. After the completion of the water bath, the sample was centrifuged at 12,000 rpm for 2 min in a high speed- centrifuge; the supernatant was aspirated and used as a DNA template for PCR amplification. Next, PCR amplification effects and SSCP (single-strand conformation polymorphism) electrophoresis were checked again; if the electrophoretic band was confirmed to be the type of band to be tested, then sequencing could proceed. DNAMAN software (version 5.2.10, Foster City, CA, USA) was used for sequence alignment, translation, and phylogenetic analyses.

### 2.8. In Situ Hybridization Analysis

The skin of the lateral part of the body of Longdong Cashmere goats was taken in October and March, embedded in wax, sectioned, and then unfolded in DEPC water and baked overnight at 37 °C. The RT-PCR products of the *KRTAP22-2* gene were cloned and sequenced, and then used to make templates for digoxigenin-labeled probes after repeated purification and denaturation of the DNA template, according to the steps of the Roche digoxin labeling kit instructions. The prepared paraffin tissue sections were dewaxed → rehydrated → proteinase K digestion → PBS rinsing and drying, pre-hybridization in a 2 × saline–sodium citrate (SSC) wet box for 1 h (42 °C). Then hybridization took place at 42 °C for 20 h, followed by a 2 × SSC → 1 × SSC → 0.25 × SSC gradient wash treatment. Then color development occurred with Nitro-Blue-Tetrazolium (NBT) for 16 h. The color rendering was terminated by autoclaved distilled water, re-stained with eosin dye, dehydrated and sealed with neutral gum, and finally examined and photographed under a light microscope.

### 2.9. Analysis of the Association between Genetic Variation and Cashmere Traits

Experimental data were processed and analyzed using SPSS V24.0 (IBM, Armonk, NY, USA) software. To ensure the accuracy of the data analysis, general liner mixed-effect models (GLMMs) were used to correlate individuals with *KRTAP22-2* genotype frequencies and allele frequencies greater than 5% with cashmere traits (fleece yield, cashmere mean fiber diameter, and crimped fiber length). The ANOVA results showed that gender and sire affected all fiber traits (*p* < 0.05); therefore, in the GLMMs model, they were designated as fixed and random factors.

## 3. Results

### 3.1. Identification of KRTAP22-2 in the Goat Genome

A BLAST homology search of the goat genome (NC_030808.1) with reference to the sheep *KRTAP22-2* gene sequence (NM_181615) revealed that a DNA sequence with 100% similarity to sheep *KRTAP22-2* was present on goat chromosome 1. Analysis of this DNA sequence revealed a 147-bp stretch of open reading frame (ORF; NC_030808.1: nt3522006_nt3522152. This ORF clustered with 13 identified goat KRTAPs and was located in the middle of different *KRTAP* families (Figure 1). Two PCR primers were designed for amplification of the gene and the amplicon size was confirmed to be 317 bp by sequencing.

The amino acid sequences of selected human, sheep, and goat KAPs were downloaded from the NCBI database. Among them, the sheep KAP22-2 was included, and a phylogenetic tree was constructed together with the amino acid sequence of the newly identified goat KAP22-2. The results showed that the newly identified goat KAP22-2 clustered together with sheep KAP22-2 with the closest genetic distance and were further distantly related to other known KAP clusters, forming a population distinct from other KAP families (Figure 2). The results confirmed the identification of KAP22-2 in goats.

### 3.2. Detection of Allelic Variation in Caprine KRTAP22-2

A total of four specific bands were detected in 348 Longdong Cashmere goats (results are shown in the Appendix A), and they formed a total of six band types either individually or in combination (Figure 3). Four alleles, named A, B, C, and D, were identified by DNA sequencing of the PCR amplification products, and these alleles comprised a total of six genotypes in the form of homozygote and heterozygotes. The gene frequencies of the four alleles A, B, C, and D were 89.7%, 7.0%, 3.0%, and 0.3%, respectively, in which allele B was the dominant allele. The frequencies of the six genotypes were AA (80.7%), AB (12.4%), AC (5.2%), AD (0.3%), BB (0.9%), and CC (0.5%).

A total of four single nucleotide polymorphic sites (SNPs) were detected in the sequenced four *KRTAP22-2* allele sequences: Two of them were located in the 5’UTR region (c.-98G/A and c.-85T/G), and the other two SNPs were located in the coding region (c.25G/C and c.122G/A), and both of them were non-synonymous mutations, which causing glycine at position 9 to become arginine (p.Gly9Arg) and arginine at position 41 to become lysine (p.Arg41Lys). In addition, a 6-bp insertion sequence (c.57_c.62 ins GTAGAT) was present in allele C, resulting in two amino acid insertions in goat *KRTAP22-2* allele C (p.19_p.20insRC) (Figure 4).

The four alleles of goat *KRTAP22-2* encode a polypeptide chain consisting of 46 or 48 amino acid residues (Figure 5). Among these polypeptide chains, tyrosine and glycine had the highest content, reaching 21.74–20.83 mol% and 15.22–14.58 mol%, respectively, moderate levels of serine and cysteine (13.04 mol% and 12.50–10.87 mol%). Other amino acids were lower, such as phenylalanine 8.70–8.33 mol%, arginine 6.52–4.17 mol%, alanine, histidine, isoleucine, leucine 4.34–4.17%, glutamic acid, methionine, asparagine, proline 2.17–2.08 mol%, and lysine 0–2.08 mol%.

### 3.3. Results of In Situ Hybridization

C1, C2, C3, and C4 are the results of *KRTAP22-2* mRNA expression in the skin of Longdong Cashmere goats during the villous growth phase and resting phase, respectively (Figure 6). The results showed that during the follicular growth phase, the hair follicle structure was intact and clear, and the cells were arranged in a tight and orderly manner, while during the resting phase of the hair follicle, some of the cells in the hair follicle died and the structure was loose. *KRTAP22-2* mRNA was strongly expressed in the cortical layer of the primary and secondary hair follicles, the inner root sheath, and the hair papillae and hair mother cells at the base of the hair follicles during the growth phase of the cashmere hair in Longdong goats, and was also found in the medullary layer of the primary hair follicles. During the resting phase, signal expression was reduced at the sites described above. No positive signal was detected for the corresponding control. In situ hybridization negative control of *KRTAP* in the skin tissue of Longdong Cashmere goats (Figure 7).

### 3.4. Association Analysis of the KRTAP22-2 Gene with Cashmere Traits

In the allele presence/absence model, among the 348 Longdong Cashmere goats used for the association analysis, although four *KRTAP22-2* alleles were detected, two allele frequencies were less than 5%, C (3.0%) and D (0.3%), so they were excluded from the association analysis and only the remaining 327 were statistically analyzed. The results showed an increase in the mean fiber diameter of cashmere in the presence of allele B (*p* < 0.05). Cashmere yield and fiber length were not obviously associated with the presence (or absence) of alleles A or B (*p* > 0.05) (Table 1).

In the association analysis of *KRTAP22-2* genotypes with cashmere traits in Longdong Cashmere goats (Table 2), BB, CC, and AD were excluded from the association analysis because their genotype frequencies were less than 5%, only AA, AB, and AC with genotype frequencies greater than 5% were analyzed. The results showed that the mean fiber diameter of goats with the AB genotype was significantly higher than that of the AA and AC genotypes. No notable effect of the *KRTAP22-2* genotype on cashmere yield and fiber length was found in Longdong Cashmere goats.

## 4. Discussion

Currently, 30 *KRTAPs* have been identified in sheep, whereas only 18 of them have been identified in goats. In this study, we applied molecular marker-assisted selection to continuously discover new genes in the *KRTAP* family, and also analyzed the possibility of using related KAPs genes for molecular marker-assisted breeding. The results show that variation in this gene significantly affects cashmere diameter. This provides theoretical data for future research and improvement of fleece quality, breeding of new Longdong Cashmere goat strains with high fleece yield and excellent fleece quality.

Using the sheep *KRTAP22-2* gene as a model, an open reading frame of 147-bp size was obtained by comparison on goat chromosome 1 using the BLAST function of GenBank. This gene clusters with other KAP genes on goat chromosome 1. This is generally consistent with the location of the *KRTAP22-2* gene on the sheep chromosome. By constructing a phylogenetic tree, we found that the sequence of *KRTAP22-2* of the goat had the highest homology with that of sheep, and the similarity of this sequence with the *KRTAP22-2* sequence of goat indicated that it represented the *KRTAP22-2* gene of goat. The goat *KRTAP22-2* gene encodes a polypeptide chain consisting of 46 or 48 amino acid residues. The highest content of tyrosine and glycine was 21.74–20.83 mol% and 15.22–14.58 mol%, respectively, in accordance with the amino acids of HGT-KAPs (containing 35–60 mol% of glycine/tyrosine) as a class of proteins [21]. Therefore, *KRTAP22-2* can be classified as HGT-KAPs, which is consistent with the classification of *KRTAP22-2* genes in sheep. In genomic DNA, any base can be mutated, so SNPs can exist in both coding and non-coding regions; however, due to genetic natural selection pressure, there are more SNPs in the non-coding regions than in the coding regions. Coding region SNPs can directly alter gene expression to affect the structure and function of the corresponding protein, ultimately leading to changes in the cashmere phenotype [22]. An amino acid change does not necessarily lead to a change in protein function. If the amino acid change does not affect the spatial structure of the protein and the amino acid is not in the active center, then the amino acid change will not have an effect on the function of the protein. However, variants in non-coding regions can also have an impact on the regulation of gene expression. The 3’UTR of eukaryotic mRNA not only regulates the stability and degradation rate of mRNA during gene expression, but also regulates the translation efficiency of mRNA [23]. Compared to the sheep *KRTAP22-2* gene, the goat *KRTAP22-2* allele A is identical to the sheep *KRTAP22-2* nucleotide sequence, whereas allele C has an insertion sequence of 6-bp length (c.57_c.62insATGTAG) and results in a 2 amino acid residue insertion (p.19_p.20insRC). The genetic variation detected in the *KRTAP22-2* gene in goat and sheep populations showed a large difference, which may be due to the different natural selection pressure on the *KRTAP22-2* gene in the two species.

The *KRATP* gene may play an important role in determining fiber structure, most of the polymorphisms found in the *KRTAP8-1* and *KRTAP7-1* genes of the Argentine llama involve changes in tyrosine and glycine residues, which have an impact on the ratio of protein specific components and consequently alter fiber properties [24]. Variation in *KRTAP7-1* was found to affect wool yield in sheep [25] and variation in *KRTAP6-3* was found to affect fiber diameter [26]; variation in sheep *KRTAP8-1* affects the mean staple length of wool fibers [27]. *KAP22-1* gene variation in sheep affects wool curvature [28]. A negative correlation between *KAP13-1* gene variation and fiber traits has been found in Changtang goat populations [29]. Variation in the *KAP6-1* gene in Egyptian Barki sheep affects wool length, diameter and yield [30]. The results of the association analysis between the goat *KRTAP22-2* gene and cashmere traits in Longdong Cashmere goats showed that in the presence/absence model, the mean fiber diameter of cashmere increased by 2.2% in the presence of allele B compared to that in the absence of B. The mean fiber diameter of goats with the AB genotype containing allele B was highly higher than that of AA and AC types (*p* = 0.009). It is suggested that the presence of allele B is associated with an increase in MFD and therefore selection of goat individuals without allele B may facilitate the production of finer cashmere fibers. However, considering that this association was detected only in allele B and not in allele A, the reason for this difference could be due to the presence of a nonsynonymous mutation in the coding region (c.25G/C) resulting in a change of glycine to arginine (p.Gly9Arg). Although the role of glycine in HGT-KAPs has not been determined, glycine is the smallest amino acid and lacks a side chain, which may make HGT-KAPs more flexible and thus better able to form tight structures with IF proteins [31]. It has been reported that HGT-KAPs are expressed soon after IF protein synthesis [32,33]. The deletion of glycine in goat *KRTAP22-2* allele B may affect the *KAP22-2* protein’s ability to regulate the stacking density and helix angle of IFs and its interaction with adjacent IFs and KAP, thereby affecting the average fiber diameter of cashmere. Notably, an insertion sequence of 6 bp in length was detected in allele C (c.57_c.62insGTAGAT), but the lack of correlation between *KRTAP22-2* allele C and cashmere traits in this study suggests that the 6 bp insertion sequence in allele C is unlikely to affect cashmere fiber diameter and cashmere weight.

In humans, all the KAP genes that have been identified to date are expressed in the hair follicle, except for families 16, 22, 25, and 27 [10,11,34]. The expression of *KRTAP22-2* in the growing and resting phases of cashmere fibers was investigated by in situ hybridization assays. *KRTAP22-2* showed a strong expression signal in the medulla of primary fiber follicles, suggesting that this gene was a component of the fiber medulla and had important effects on the nature of fiber. The positive expression signal of *KRTAP22-2* was also detected in the intrafollicular root sheath in this study, suggesting that *KRTAP22-2* may be related to the growth of intrafollicular root sheaths during the flourishing phase, and is involved in the formation of intrafollicular root sheaths, which play an important role in the growth, development and quality of goat fleece [35]. It was further observed that *KRTAP22-2* was expressed in the hair papillae and hair maternal cells at the base of the hair follicle during the growth phase, and was not expressed or weakly expressed in the above sites during the resting phase. This is in general agreement with KAP1 family members in sheep [36], and *KAP1.5*, *KAP1.6*, *KAP1.7*, *KAP10*, and *KAP12* expression sites in human hair follicles [37]. During the growth phase of fiber follicles, especially in September–October when the differentiation of primary and secondary follicle cells is accelerated, the villi are in a rapid growth phase and the expression of genes for keratin synthesis is most active. Thus, the expression site and results of *KRTAP22-2* were consistent with the expected. In the human genome, members of the KRTAP gene family exhibit similar hair follicle expression patterns, except for HS-KAPs and HGT-KAPs on chromosome 21 and *KRTAP17-1* on chromosome 17 [38,39,40]. It may be determined by the characteristics of the chromosomal regions where these gene families are located.

In summary, a nonsynonymous mutation (c.25G/C) in the coding region of the goat *KRTAP22-2* gene may affect the structure or expression of KAP22-2 protein or its interaction with IF protein, thereby affecting the average fiber diameter trait of cashmere. However, this association may also be due to its association with other genes on the chromosome 1 of the goat. Since the average fiber diameter of cashmere is characterized by high heritability [41,42], the newly identified *KRTAP22-2* gene might play a greater role in goat breeding for reducing cashmere fiber diameter.

## 5. Conclusions

This study identifies a new gene encoding a KAP in goats and reveals the association of variation in this gene with cashmere traits. It was found that variation in the goat *KRTAP22-2* resulted in changes in cashmere fiber diameter. The results of this experiment could be used in breeding programs related to the improvement of cashmere diameter.

## Figures and Tables

**Figure 1 animals-13-02806-f001:**
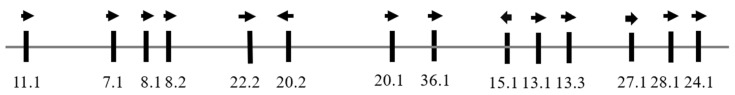
Location of *KRTAP22-2* and other *KRTAPs* on caprine chromosome 1. Vertical bars, arrows and numbers indicate the position of the different *KRTAPs*, the direction of transcription and the name of the *KRTAPs*, respectively (e.g., 20.1 represents *KRTAP20-1*).

**Figure 2 animals-13-02806-f002:**
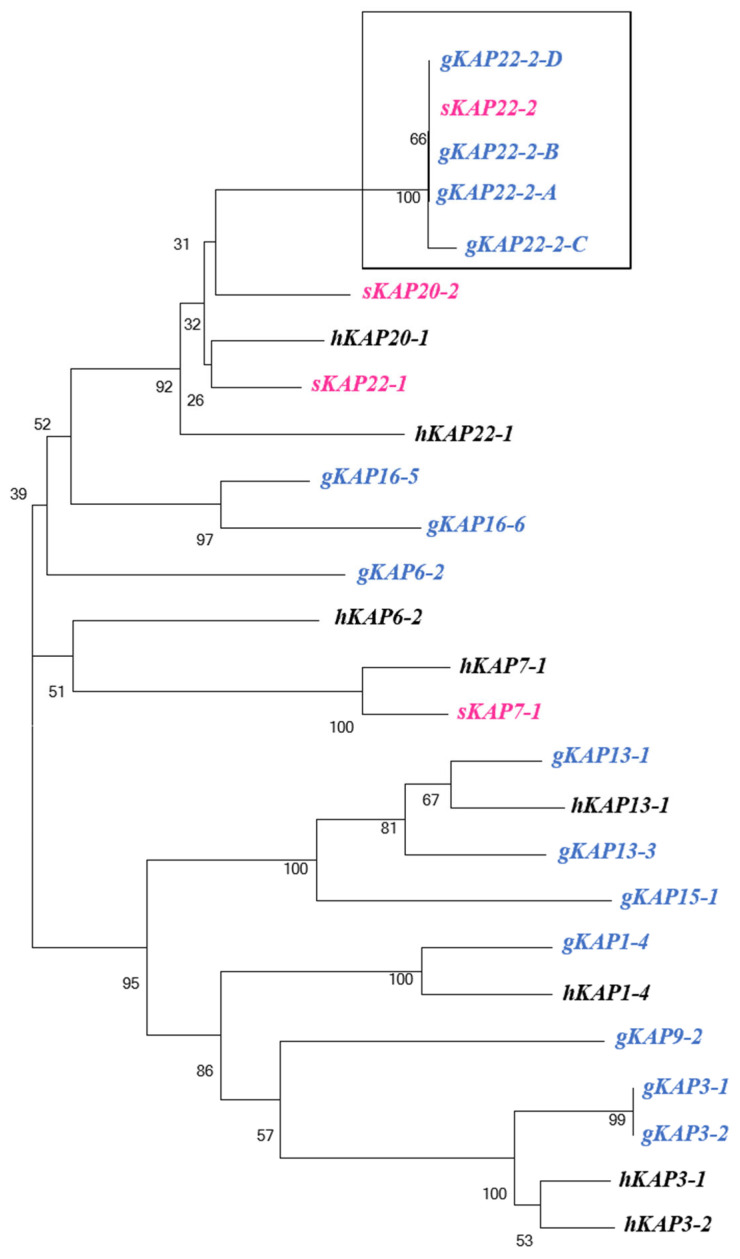
The newly discovered KAP22-2 sequence in goats and other KAPs found in goats, sheep, and humans. Phylogenetic tree constructed. Goats are denoted by the prefix “g” (blue), sheep by the prefix “s” (pink), and humans by the prefix “h” (black). The caprine KAP22-2 sequences identified in the study are shown in a box.

**Figure 3 animals-13-02806-f003:**
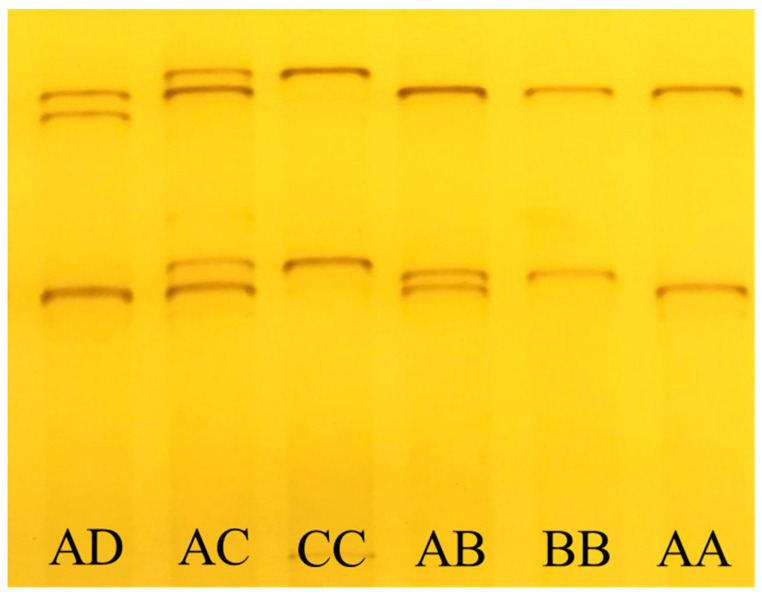
PCR-SSCP assay results for the *KRTAP22-2* gene in Longdong Cashmere goats with four specific bands detected.

**Figure 4 animals-13-02806-f004:**
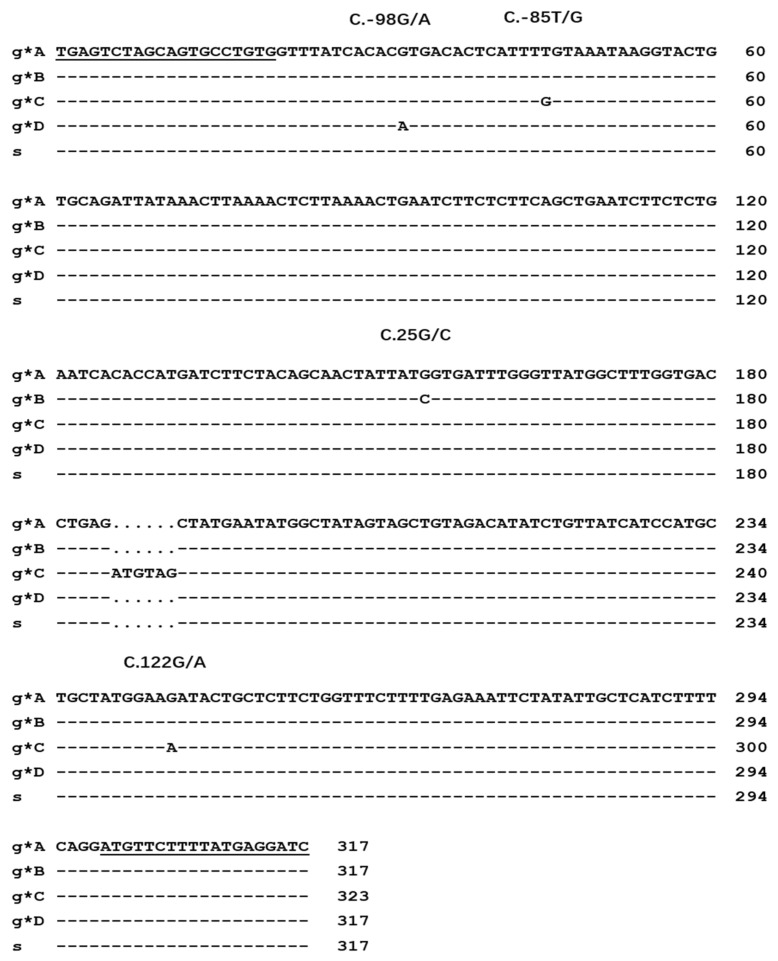
Sequence alignment of the goat and sheep *KRTAP22-2*. Goat sequences are labeled with the prefix “g” and sheep sequences with the prefix “s”; nucleotide sequences that are identical to the first row of sequences are indicated by dashes; and dots indicate the absence of nucleotides; underlining indicates primer sequences.

**Figure 5 animals-13-02806-f005:**
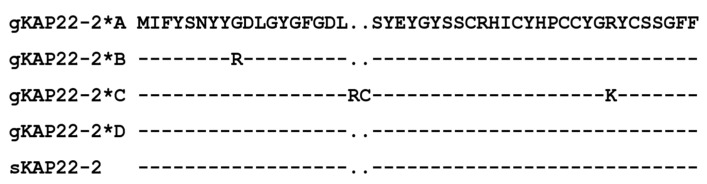
Comparison of predicted amino acid sequences of goat and sheep *KRTAP22-2* variants. Dashes indicate amino acids identical to the sequence above, and dots indicate amino acid deletions. Goat and sheep sequences are prefixed with “g” and “s”, respectively.

**Figure 6 animals-13-02806-f006:**
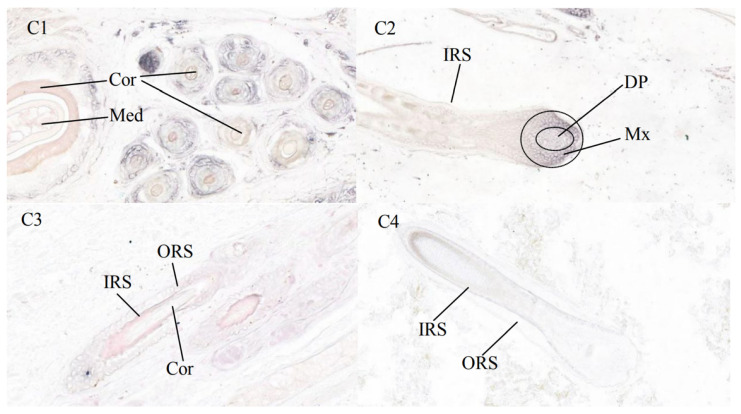
The expression results of *KRTAP22-2* mRNA on Longdong Cashmere goat skin. C1 is a growing season transverse cut (30×), C2 is a growing season longitudinal cut (30×), C3 is a resting period longitudinal cut (30×), C4 is a resting period longitudinal cut (30×). Among them, IRS: inner root sheath; ORS: outer root sheath; Med: medulla; Cor: cortical layer; DP: dermal papilla cells; Mx: hairy parent cells.

**Figure 7 animals-13-02806-f007:**
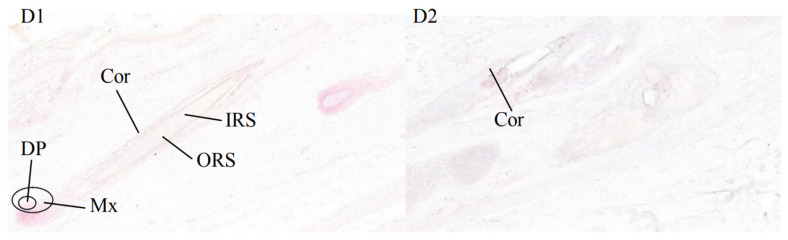
Expression results of *KRTAP22-2* mRNA telogen phase on Longdong Cashmere goat skin. D1 is a resting period longitudinal cut (30×), D2 is a resting period longitudinal cut (30×).

**Table 1 animals-13-02806-t001:** Allele presence/absence model and cashmere trait association analysis.

Projict	Variant	Absent		Present		*p* Value
		Mean ± SE		Mean ± SE		
Cashmere yield(g)	A	410 ± 3.3	3	415 ± 11.4	324	0.733
	B	410 ± 3.1	281	420 ± 15.1	46	0.523
Mean fiber diameter(µm)	A	13.6 ± 0.03	3	13.6 ± 0.11	324	0.899
	**B**	**13.5** **± 0.03**	**281**	**13.8 ± 0.14**	**46**	**0.041**
Cashmere fiber length(cm)	A	4.2 ± 0.04	3	4.1 ± 0.12	324	0.254
	B	4.2 ± 0.03	281	4.4 ± 0.16	46	0.149

Note: Estimated marginal means and standard errors were derived from a general linear mixed effects model with “gender” and “parentage” as fixed and random factors, respectively. (*p* < 0.05) marked in bold.

**Table 2 animals-13-02806-t002:** Association analysis between KRTAP22-2 genotype with cashmere traits in Longdong Cashmere goats.

Projict	Mean ± SE			*p* Value
	AA (n = 281)	AB (n = 43)	AC (n = 18)	
Cashmere yield(g)	405.7 ± 4.5	434.2 ± 17.3	444.1 ± 22.3	0.134
Mean fiber diameter(µm)	**13.5 ± 0.04 ^b^**	**14.1 ± 0.21 ^a^**	**13.6 ± 0.16 ^b^**	**0.009**
Crimped fiber length(cm)	4.2 ± 0.05	4.2 ± 0.18	4.4 ± 0.24	0.150

Note: The numbers with different superscript letters were significantly (*p* < 0.01) different and represented in bolded.

## Data Availability

The authors affirm that all data necessary for confirming the conclusions of the article are present within the article, figures, and tables.

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
