# Peer review of "Identification of the Keratin-Associated Protein 22-2 Gene in the Capra hircus and Association of Its Variation with Cashmere Traits"

_animals, 2023, doi:10.3390/ani13172806_

Round 1
Reviewer 1 Report
Dear Authors,
Zhanzhao Chen et al. provide a very interesting investigation on genetic variation in Keratin-associated protein (KAP) gene, and its association with Cashmere quality in Chinese goats. The authors used comparative data from sheep (KRTAP22-2 gene) and identified a new HGT-KAP gene (KRTAP22-2) in Chinese goats, which was previously reported in human, sheep, and mouse but novel to this study as never was discovered in goats. Through various molecular biology techniques (PCR, in situ hybridisation, etc), they identified four alleles (A-D), including a total of four single non-synonymous SNPs. In addition, specific to allele C they found a 6-bp insertion. For data analysis, the excluded genotypes with a gene frequency of less than 5% (i.e., BB, CC and AD), and used only genotypes AA, AB and AC for downstream analysis. Their results demonstrate that MFD of goat fleece was significantly higher in goats with AB genotype than in the AA and AC. The results presented in this manuscript has potential in contributing to the knowledge of the Caprine research community, and future breeding programs, in terms of using goat KRTAP22-2 variants as a molecular marker for improving the cashmere quality in this goat breed. However, more genomic studies on other goat populations would be crucial prior to using this marker for breeding programs.
The manuscript is written clearly, the methodologies are adequately explained, and the obtained results support the conclusion of the study. However, the presentation of the results can be improved (written and graphical). The discussion in the current form reads more like the “Results section”, so I strongly advise to re-write in the form of putting the results in the context of such studies in other species (human, sheep, etc).
I very much enjoyed reading this paper, and I have some minor comments as below:
Line 11: remove “of cashmere” in the sentences below:
It has been shown that the keratin- 10 associated protein composition and genetic diversity of cashmere are closely related to its cashmere 11 fiber quality.
Line 19: … KRTAP22-2 gene and cashmere [production] trait.
Line 19: …. highly significantly higher ….
Line 26: aligned in [to] the goat genome, …
Line 31: A total of four [non-synonymous] single nucleotide polymorphic sites …
Line 33-34: … region, and all of them were non-synonymous mutations.
Line 38-39: Do you mean goat or sheep here “ …AB genotype sheep than in the AA and 38 AC genotype sheep.”
Line 42-43: please use few more informative keywords, and also use the scientific name of the goat (Capra hircus)
Line 46: …. mean fiber diameter (MFD) …..
Line 47: …., and the most important way in which cash- 47 mere demonstrates its economic value [1].
This part of the phrase is redundant, please remove.
Line 51-52: …. properties of the hair and 51 wool fibers [references??].
Line 82: … the cashmere [production] trait
Line 79-83: there are present and past verbs in this phrase, please re-write and uniformly use verb tenses.
Line 95-97: please re-phrase, the sentences make no sense!
Line 97: Collected fiber samples collected ….
Line 97-98: …mean fiber diameter MFD…
Line 103-104: ….no genetic correlation between them, …
Please explain how did you assess the genetic relatedness between individuals?
Line 109-115: please add information of the length of this homologous gene in goat.
In addition, from the paragraph below, I have an understanding that there are couple of sequences, is that the case, or you retrieved one long sequence with 100% similarity to sheep gene.
“These sequences were used to design PCR primers to amplify the entire coding region of the gene 114 from goat DNA.”
Line 138: replace “2.6. Sequencing of Allelic Variants and Sequence Analysis” with
2.6. Sequencing and Analysis of Allelic Variants
Line 151: make a space between “ )” and “electrophoresis”.
conformation polymorphism)electrophoresis
Line 193-194: this sentence is unclear “Download the amino acid sequences of selected KAPs that have been identified in 193 humans, sheep and goats from the NCBI database. “
Figure 202-205: in figure 2, please use different colors for human, sheep and goat with a inset representative of the colors for each species. In the current form, with prefix, is visually rather hard to clearly see the comparison.
Line 242: 3.3. Results of KRTAP22-2 In Situ Hybridization
Line 279-282: Based on the information provided in the paragraph below, can you please explain if any particular cashmere-related phenotype can be associated with goats with genotypes BB, CC and AD?
“In the association analysis of KRTAP22-2 genotypes with cashmere traits in Long- 279 dong cashmere goats (Table 2), BB, CC and AD were excluded from the association anal- 280 ysis because their genotype frequencies were less than 5%, only AA, AB and AC with gen- 281 otype frequencies greater than 5% were analyzed.”
Best Wishes,
Moderate editing of English language required.
Author Response
感谢您的耐心等待,有关文章建议的回复在附件中,再次感谢。

Reviewer 2 Report
ANIMALS - MDPI
Referee’s Evaluation Report
MANUSCRIPT IDENTIFICATION: animals-2543791
Identification of the goat keratin-related protein 22-2 gene and its variant association analysis with Cashmere traits in Chinese goat breeds
(ORIGINAL ARTICLE)
Comments to Authors/Editor:
The paper of Chen & colleagues aimed to identify a sequence encoding a putative caprine KRTAP22-2 and applied DNA sequencing and other techniques to reveal the chromosomal location, variation, and amino acid composition of such gene. Besides the authors analyzed the relationship between the Cashmere trait and the variation of the gene in Longdong Cashmere goats. Moreover, thru in situ hybridization the expression-location of the KRTAP22-2 gene in the fiber follicles of goats was assessed while explored the possible effect of KRTAP gene on some physicochemical properties of cashmere from a histomorphology standpoint. This manuscript falls within the scope of ANIMALS, being ad-hoc for the Special Issue. The manuscript is sufficiently informative for the replication of the study. In general, the organization of the experiment seems to be well designed, yet, the English quality, grammar, and sentence structure can be improved. Simple Summary and Abstract are clear; L39; incomplete sentence; correct accordingly. The authors MUST clarify some numbers, especially to include the number of animals evaluated as well as the geographical origin of the breed (i.e., NL, WL), how the goats were sampled and what tissues were collected: Regarding the fiber, from what part of the body the fiber samples were taken, how many samples by place, and by animal within the breed were studied. Why this breed was chosen, and no other breeds were considered???, Please contextualize. These issues must be carefully addressed-clarified in both sections, especially in the Abstract section. The authors must simplify the information because of the large set of response variables, and only include those response variables mainly involved with the fiber quality. Certainly, the authors must select some key variables to include the observed value along with the probability level. The Introduction section is generally OK; the authors must contextualize the social, economic, and productive importance of the Longdong Cashmere goats within China´s Agriculture subsector; please include some economic and census information. While the objectives of the study were included in the Introduction section, the working hypothesis of the study was never proposed; this is a must. In the Material & Methods section, I do strongly recommend starting with the general information; besides the institutional approval of the study, the authors must state if the study follows any international guide for the use of animals in research; this is a must. Thereafter, as subsection 2.1, include information regarding “Location, environmental conditions and animals”, then, 2.2. Blood and Cashmere sampling of goats. L95-96; please re-write. L102, why the authors also sampled males from the Liaoning breed???; they never mention about such issue in the objectives, of the Introduction section. L110-111, uncomplete sentence. In the M&M section, I strongly recommend including a figure with the actual experimental protocol across time (i.e., a timeline of actions); this is a must. Reagents, standards, and methods used are relevant and in accordance with the objectives of the study. Also, all the treatment, sampling techniques, laboratory methods, as well as response variables considered in the experiment are detailed and accurate, while in agreement with the objectives of the study. The experimental design was not explained, neither the statistical models were described well enough for the reader to understand how the experiment was carried out. Regarding the Results section, the novelty value of the results is reasonable. In this section the authors must avoid the use of both the word “significant” and the “probability value itself”; it is a pleonasm. (i.e., L283-284). Results were shown in 7 Figures and 1 Table; the titles must be rewritten; the authors must include the number of replicates and mention if they collected repeated samples across time. The titles of tables must stand by themselves; titles must be rewritten. Regarding the Discussion section, at the beginning of the Discussion, I do strongly suggest to initiate this section including the working hypothesis of the study. Authors must define if, with the obtained results, such hypothesis is rejected or non-rejected. For this reason, the authors must include the working hypothesis prior to the objectives in the Introduction section. In addition, the authors must follow the same order in this section according to that proposed in the Results section. The authors must link, in a logical fashion, their main findings along with the discussion section, to compare & to discuss and, thereafter, be able to propose some possible explanations for such specific outcomes, considering to previous similar studies from the scientific literature. In general, the authors made an accurate interpretation of the main findings. The authors must focus their main findings and confront them with respect to the scientific literature in a logical and focused fashion. The authors repeat some ideas already presented in the Results section; correct accordingly. The main outcomes of the study were soundly presented. The list of references cited in the manuscript is proper. This is a very interesting while innovative study. Yet, the authors must improve the clarity and logical arrangement of the observed results, especially in the Discussion section. The authors must align the conclusions regarding the working hypothesis as well as the scientific question they try solving; nothing else, just that. All the commented issues and requests should be clearly addressed by the authors; at this point, and based on the above comments, my pronouncement is that this manuscript requires moderate adjustments prior final acceptance.
Moderate grammar adjustments are needed.
Author Response
感谢您的耐心等待,有关文章建议的回复在附件中,再次感谢。
